# Pelvic organ prolapse: The lived experience

**Louise Carroll** [1,2,3]*, **Cliona O' Sullivan**[1], **Catherine Doody**[1,2], **Carla Perrotta**[1],
**Brona Fullen**[1,2]

**1** University College Dublin School of Public Health, Physiotherapy and Sports Science, Dublin, Ireland,
**2** University College Dublin Centre for Translational Pain Research, Dublin, Ireland, **3** Tipperary University
Hospital, Clonmel, County Tipperary, Ireland

* maria-louise.carroll@ucdconnect.ie

doi.org/10.1371/journal.pone.0276788

Palermo: Universita degli Studi di Palermo, ITALY

**Data Availability Statement:** All relevant data are
within the paper.

**Funding:** LC Grant number not applicable. This
work is funded by the UCD Centre for Translational
Pain Research (https://www.ucd.ie/ctpr/) The
funders had no role in study design, data collection

## Abstract

### Background

Up to 50% of women will develop pelvic organ prolapse (POP) over their lifetime. Symptoms
include pain, bulge, urinary, bowel and sexual symptoms affecting all aspects of a woman's
life. This study explores the lived experience of women with POP.

### Methodology

A qualitative study was undertaken. Following institutional ethical approval women from an
online peer support group (n = 930 members) were recruited to participate in semi-struc-
tured interviews. Inclusion criteria stipulated women (> 18years), pre-menopausal, at least
one-year post-partum, diagnosed with POP and aware of their diagnosis. Semi-structured
interviews were undertaken with a clinician specialising in pelvic health. A battery of ques-
tions was designed to elicit discussion on their experience of being diagnosed with POP and
its impact on daily life and relationships. Interviews were carried out via Zoom, recorded and
transcribed. Thematic analysis was undertaken.

### Findings

Fourteen women (32–41 years), para 1–3 participated. All had at least one vaginal birth;
three had vacuum, four had forceps operative births. All had Grade 1–3 POP. Interviews
lasted 40–100 minutes. Three core themes with subthemes were identified; biological/physi-
cal, psychological and social. Women were particularly affected in terms of sport and exer-
cise participation, their own perceptions of their ability as mothers and fear of their condition
worsening. They described societal attitudes, reporting stigma around POP and women's
pelvic health in general, expectations placed on women to put up with their symptoms and
an idealised perception of new motherhood.

### Conclusions

The impact of POP from a biopsychosocial perspective reflects other chronic conditions.
Prevention, early education and supports for developing strong self-management
approaches would be beneficial for long term management of this condition.

and analysis, decision to publish, or preparation of the manuscript.

**Competing interests:** The authors have declared that no competing interests exist.

## Introduction

Female Pelvic organ prolapse (POP) is defined by the International Urogynecological Association (IUGA) and International Continence Society (ICS) as a departure from normal sensation, structure, or function, experienced by the woman in reference to the position of her pelvic organs [1].

Prevalence rates of POP have been reported at 1–65% globally [2]. There is a dearth of prevalence data for POP and most existing prevalence data is based on symptoms rather than physical examination [2]. When defined by symptoms, POP has a prevalence rate of 3–6%, and up to 50% when based upon vaginal examination [3].

It is unclear how likely it is that asymptomatic POP worsens and becomes more symptomatic over time. Wu et al (2014) [4] reported that among women with symptomatic POP, those between the ages of 20–29 years account for 6% of those with POP, women aged 50–59 years account for 31% and about 50% of women with POP are aged 80 years or older. It has been reported that between 19–48% regression of stage 1 or 2 prolapse occurs without intervention over three to eight years [5].

A broad range of symptoms are commonly associated with POP, including sensation of a bulge or fullness in the vagina, feeling "like a tampon is falling out," pelvic pressure, groin pain, low back pain, painful intercourse, difficult bowel movements, urinary or faecal incontinence, sexual dysfunction [6], difficulty achieving orgasm and lack of vaginal sensation [7]. Vij et al [8] also established that 32% of women with POP show signs of central sensitisation (increased responsiveness of nociceptive neurons in the central nervous system to their normal or subthreshold input [9]) as measured by the Central Sensitization Inventory (CSI) [10].

These symptoms have a significant impact on women's lives: impinging on sexual health, restricting daily and sports activities and affecting women's ability to fulfil everyday parental duties [11]. Women report negative changes in body image [12, 13], annoyance, frustration, and irritation, unhappiness, depression, anxiety and sadness [14]. In terms of treatment and management they also report feeling they weren't listened to by healthcare professionals (HCP), given information, or counselled on the options for treatment of their symptoms [6, 15].

Pelvic organ prolapse research to date has been primarily quantitative and has focused heavily on surgical management and prevalence studies, with less emphasis on the experiences of women living with and seeking care for this condition. However, there is now a greater recognition of the usefulness of qualitative research for detailed exploration of the patient perspective, an area often inaccessible in other research methods [16].

Internationally, qualitative research on women's experience of POP has focused on specific aspects; symptom bother and its impact on daily life and emotions [6, 11, 14], body image [13], seeking care for POP [6, 15] or women's perspectives on available treatment options [15]. In general the overall lived experience of younger pre-menopausal women diagnosed with and seeking care for POP is limited. Given the high prevalence rate and negative impact of POP on women's quality of life the lack of qualitative research represents a significant deficit. Hence establishing this perspective is essential to provide a holistic picture of the problem so that effective preventative and early management strategies can be implemented.

## Methodology

A qualitative research methodology was employed using the EQUATOR standards for reporting research [17]. An interview guide, informed by the contemporary literature, the principal investigator's (PI) expertise and the study aims was developed and piloted (Table 1). Participants were recruited, semi-structured zoom recorded interviews conducted and thematic

**Table 1. Structured interview guide.**

| Question | |
|---|---|
| 1 | When did you learn you had a prolapse? |
| 2 | What stage is your prolapse? |
| 3 | Who diagnosed it? |
| 4 | How did you feel about it at the time? |
| 5 | Do you think your symptoms have an impact on your mood? |
| 6 | Do they have an impact on your sleep quality or vice versa? |
| 7 | Do your prolapse symptoms have an impact on your home/family life? If so, how? |
| 8 | Do your prolapse symptoms have an impact on your relationships? |
| 9 | Do they affect your work life? |
| 10 | Have you noticed any specific triggers for your symptoms? If so, what? |
| 11 | Have you noticed anything that helps your prolapse symptoms? If so, what? |
| 12 | Are there things that your prolapse stops you from doing? |
| 13 | Do your prolapse symptoms have an impact on how you feel about yourself? |

analysis undertaken. Ethics approval was obtained from by University College Dublin's Human Research Ethics Committee (LS-21-01-Carroll-Ful).

## Recruitment

Women with POP registered to an online support group (n = 930 members) were invited to participate by an advertisement on the support group website outlining the study and inclusion criteria. Inclusion criteria were adult women (>18 years), premenopausal with POP diagnosed and staged by a HCP, were at least one year post-partum, not currently pregnant and able and willing to give informed consent. Sampling was undertaken to ensure range of POP stages (I-3). Stage 4 usually requires surgical intervention; therefore this cohort were not included in the sample.

Women who were interested in participating were invited to contact the PI through contact details on the recruitment advertisement. They were screened for eligibility by the PI via zoom, the study was explained and the information leaflet and interview guide (Table 1) forwarded via e-mail. Potential participants were given a period of one week to consider participating in the research after which they re-contacted the PI, and once any questions were answered and the informed consent form received, they were accepted into the study.

A participant resource leaflet, and the battery of questionnaires were forwarded by return email to be completed electronically using Google forms. They comprised:

1. Demographic Questionnaire
   This included questions relating to participant demographics, obstetric and gynaecologic related characteristics (age, parity, birth mode(s), time since POP diagnosis, POP stage).

2. Validated Questionnaires

   i. The Central Sensitization Inventory
      This is a self-report outcome measure [10, 18] designed to identify people who have symptoms related to CS or central sensitivity syndromes (CSS). The CSI has been shown to have excellent test-retest reliability, internal consistency and construct validity and good responsiveness and interpretability [19].

   ii. Pelvic Organ Prolapse Quality of Life Questionnaire
      This questionnaire assesses the severity of symptoms and their impact on the QOL in

women with POP. It consists of 20 items, grouped into 9 domains. Higher scores indicate lower QOL. It also has been shown to have good test-retest reliability, internal consistency and construct validity [20].

## Semi-structured interview protocol

A realist phenomenological approach was taken to investigate the lived experience of women with POP. This approach [21] has two foundations; critical realism which theorizes that the 'real' cannot be observed and exists independent of human perceptions [22], and phenomenology, which focuses on exploring the study of an experience from the perspective of an individual [23].

Data collection using Zoom was chosen for several reasons; geographic location of the participants (more convenient to interview online than in person), COVID 19 restrictions (due to several lockdowns and restrictions on travel), lower chance of public encounters between interviewer and participants (due to the sensitivity of the subject matter) this was felt to be important and has been shown in research to be an important consideration for participants [24].

Interviews were conducted by the PI, a senior physiotherapist with 12 years of expertise in women's health. Follow-up questions and comments were also used to allow participants to elaborate on their answers. To standardise the semi-structured interviews and reduce response bias, a statement was read out at the beginning of each session. The participants were reminded of the aim of the study, that information was confidential and they would not be identified in any way. Interviews were carried out at any time of the day or day of the week convenient to participants, in their homes during Covid 19 lockdowns.

## Data analysis

Participant demographics and completed questionnaire scores were compiled and scored. The Zoom programme recorded and automatically transcribed the interviews which were then reviewed for accuracy, any identifying information removed, and returned to the individual participants to confirm that their interview was accurately reflected. Once completed a database was formatted by anonymising data by identifying the participant by number [1–14], and the transcripts were line-numbered to allow for coding of data and citation purposes, e.g participant number (T1), page 5 (P5), line 3 (L3).

Thematic analysis was used to analyse the data as outlined by Braun and Clarke: (familiarising with the data, generating initial codes, searching for themes, reviewing themes, defining themes and writing up) [25].

The interviews were read and re-read to ensure a 'thorough feel for the meaning' of the data. To facilitate reduction of the data, an iterative process was undertaken where a preliminary coding scheme was developed, revised and expanded to include other primary recurring codes as they emerged until data saturation was achieved (LC) and was applied to the whole database. On reviewing the codes patterns were identified and themes developed. Refinement of the themes and development of sub themes with a second researcher (BF) resulted in three main themes emerging from the analysis.

## Trustworthiness of findings

The coding system was examined for reliability which establishes trustworthiness, credibility, transferability, and confirmability of the data [26]. The final coding system was applied to a random sample of the transcript by the PI (LC) and a second researcher (BMF) independently

and inter-rater reliability, by calculating percentage agreement was established at 89%. Intra-rater reliability was established by the PI (LC) who coded and re-coded the transcripts on two separate days (94%).

Thick description of the phenomenon was used to allow transferability judgements by the reader and chain of evidence was maintained by coding and line numbering the transcripts for citation purposes. The final template was applied to a random sample of the transcripts database.

## Results

Semi-structured interviews were conducted with 14 women diagnosed with POP. The length of the interviews ranged from 40 to 100 minutes.

### Participant demographics

All female participants were aged between 32–41 years with a mean age of 36.79±3.3 years. Mean parity was 2±0.5. Seven women (50%) reported having had an instrumental (vacuum/forceps) birth with one reported incidence of severe perineal trauma (SPD) (3rd degree tear). Five women reported having an episiotomy. Women with a range of POP stages participated: Stage I (n = 5), Stage II (n = 6) with Stage III (n = 3) women. Participant demographics are summarised in Table 2 (below).

**Table 2. Participant demographics.**

| Participant No. | Age Years | Births (Number) | Birth Type | POP Type & Stage (dominant compartment) | Associated Pelvic Floor Symptoms |
|---|---|---|---|---|---|
| 1 | 40 | 2 | 1st birth: forceps, epis, 3rd degree tear<br>2nd birth: c-section | Stage I anterior wall | SUI |
| 2 | 38 | 2 | Vaginal, epis on both | Stage I anterior wall | Urinary Frequency, Dyspareunia |
| 3 | 41 | 2 | Vaginal | Stage III posterior wall | SUI |
| 4** | 39 | 2 | Vaginal, IOL, forceps | Stage III anterior & apical | Recurrent UTIs |
| 5 | 39 | 2 | 1st birth: vacuum, forceps, epis<br>2nd birth: SVD | Stage I unsure of involved compartment | |
| 6 | 35 | 2 | Vaginal | Stage II anterior wall | Incomplete bladder emptying |
| 7 | 38 | 3 | 1st birth: IOL, forceps, epis<br>Subsequent 2 births: vaginal | Stage I anterior, posterior & apical | Pelvic pain |
| 8 | 37 | 2 | Vaginal | Stage III posterior wall | Bowel urgency with running |
| 9 | 39 | 2 | Vaginal | Stage II anterior wall | Urinary urgency |
| 10* | 33 | 2 | Vaginal | Stage II anterior wall | SUI |
| 11* | 32 | 2 | 1st birth: vacuum, epis<br>2nd birth: vaginal | Stage I posterior wall | Difficulty emptying bowel |
| 12 | 36 | 3 | Vaginal | Stage II anterior wall | SUI |
| 13 | 34 | 1 | Vacuum | Stage III anterior wall | |
| 14* | 41 | 2 | 1st birth: vacuum<br>2nd birth: elective c-section | Stage II anterior and posterior wall | Difficulty emptying bowel |

CSI = Central Sensitization Inventory, epis = Episiotomy, IOL = Induction of Labour, SVD = Spontaneous Vaginal Delivery, SUI = Stress Urinary Incontinence, UTI = Urinary Tract Infection.

\* = pessary use

\*\* = pessary use (unsuccessful)

## Validated questionnaires

(i) Central Sensitisation Inventory

Scores ranged from 22–65 out of 100. Participants were categorised as having mild (n = 4), moderate (n = 1), severe (n = 3), and extreme sensitisation (n = 4). Only two were categorised as subclinical [18].

Demographics and individual CSI scores are summarised in Table 2.

(ii) Pelvic Organ Prolapse Quality of Life Questionnaire

All 14 participants returned completed questionnaires. Four women failed to complete the general health perception domain of the P-QOL. Of the 10 women 50% (n = 5) rated their general health as good, 30%, (n = 3) as fair, and 20%, (n = 2) as very good. No participant rated their health in the poor or very poor categories. Co-morbidities are summarised in Table 3. Quality of life was affected in all domains of the P-QOL. Despite the relatively high scores in all domains (higher scores = lower QOL), severity measures were relatively low. All of the women reported symptoms of bulge, heaviness or pelvic discomfort commonly associated

**Table 3. Participant P-QOL & CSI scores.**

| Participant Number | General Health Perception | Prolapse Impact | Role Limitations | Physical Limitations | Social Limitations | Personal Relationships | Emotions | Sleep/ Energy | Severity Measures | Co-morbidities | CSI Score |
|---|---|---|---|---|---|---|---|---|---|---|---|
| P1 | | 66.66 | 33.33 | 16.66 | 0 | 55.55 | 44.44 | 66.66 | 25 | Neck injury Migraine/tension headaches | 61 |
| P2 | Very good 0 | 33.33 | 33.33 | 50 | 0 | 44.44 | 44.44 | 50 | 25 | | 30 |
| P3 | Fair 50 | 66.66 | 33.33 | 66.66 | 16.66 | 77.77 | 88.88 | 66.66 | 16.66 | IBS | 35 |
| | | | | | | | | | | Anxiety/panic attacks | |
| P4 | Fair 50 | 100 | 66.66 | 66.66 | 50 | 88.88 | 77.77 | 83.33 | 50 | | 65 |
| P5 | | 100 | 100 | 50 | 33.33 | 88.88 | 44.44 | 66.66 | 25 | Migraine/tension headaches | 59 |
| P6 | Good 25 | 66.66 | 66.66 | 33.33 | 16.66 | 66.66 | 33.33 | 50 | 33.33 | | 36 |
| P7 | Very good 0 | 66.66 | 33.33 | 33.33 | 0 | 55.55 | 100 | 66.66 | 16.66 | IBS | 50 |
| P8 | Good 25 | 66.66 | 0 | 16.66 | 0 | 0 | 11.11 | 16.66 | 0 | | 29 |
| P9 | | 66.66 | 50 | 50 | 16.66 | 55.55 | 44.44 | 0 | 25 | | 22 |
| P10 | Good 25 | 100 | 83.33 | 83.33 | 50 | 44.44 | 22.22 | 83.33 | 25 | Interstitial Cystitis | 48 |
| | | | | | | | | | | TMJ disorder | |
| P11 | Good 25 | 66.66 | 50 | 33.33 | 0 | 44.44 | 44.44 | 50 | 33.33 | Autoimmune condition, IBS | 36 |
| P12 | Fair 50 | 33.33 | 33.33 | 50 | 0 | 44.44 | 11.11 | 16.66 | 41.66 | IBS | 50 |
| P13 | Good 25 | 100 | 83.33 | 100 | 66.66 | 100 | 100 | 100 | 16.66 | | 60 |
| P14 | Good 25 | 100 | 66.66 | 83.33 | 16.66 | 88.88 | 100 | 83.33 | 33.33 | Breast cancer, depression | 62 |

Scores (P-QOL is a POP-specific quality of life scale with 9 domains. There is a four-point scoring system for each item in each domain. Scores in each domain range between 0 and 100. A high total score indicates a greater impairment of quality of life, while a low total score indicates a good quality of life. The CSI consists of two parts. Part A includes 25 questions related to common CSS symptoms. There is a four-point scoring system for each question, with total CSI score ranging from 0–100. Part B determines if the patient has been diagnosed with certain CSS disorders or related disorders, such as anxiety and depression. Higher scores indicate greater central sensitization, with lower score indicating less. Recommended severity ranges are: subclinical 0 to 29; mild 30 to 39; moderate 40 to 49; severe 50 to 59; extreme 60 to 100. A CSI score of >40-points is clinically significant, providing both good sensitivity and specificity for the presence of CSS.)

**Table 4. Overarching themes and subthemes.**

| Overarching Themes | Biological/Physical | Psychological | Social |
|---|---|---|---|
| Subthemes | Physical Symptoms<br>Aggravating Factors<br>Self-Management<br>Sleep<br>Hormones<br>Birth Mode | Knowledge<br>Beliefs & Expectations<br>Anxiety/Worry/Fear<br>Hypervigilance<br>Fear Avoidance<br>Resilience | Role as a mother<br>Role in the Home<br>Intimacy/Sexual Life<br>The Stigma of POP<br>Societal Attitude to Women and Women's Bodies<br>Work<br>Participation in Sport & Exercise |

with POP, as well as other urinary and bowel symptoms related to their POP (see Table 2). Three of the women were using a support pessary at the time of the study; n = 1 other had undergone an unsuccessful pessary trial (see * Table 2).

## Interview themes

Three overarching themes emerged from the interviews (Table 4). The results are presented in detail under the three main (and associated) sub-themes illustrated with examples from the transcripts.

## Biological/Physical theme

This theme comprised six subthemes: (i) physical symptoms (ii) aggravating factors for the physical symptoms (iii) self-management (iv) sleep (v) the role of hormones (vi) birth mode and its contribution to prolapse

*(i) Physical symptoms.* Women described a myriad of physical symptoms they associated with their POP including a feeling of bulge, dragging, pain and a range of bladder, bowel, and sexual issues such as urgency, incontinence and dyspareunia.

"*I just felt I had no support so like it was just, just like weight, you know? Just like a dragging, hanging feeling, and then. . . you felt like you didn't have. . .the strength or something, you know?*" (T7, P2, L18)

"*I feel like my insides are going to fall out.*" (T13, P3, L34)

*(ii) Aggravating factors.* Aggravating factors for these physical symptoms were often innocuous, everyday activities including hoovering, pushing a buggy, hanging out laundry or walking to work.

"*I think it's just, it's just doing too much, being on my feet, or like, picking up anything heavy.*

Or say, like, trying to push the buggy up the hill." (T5, P10, L9)

*(iii) Self-management.* All participants were mothers of young children and struggled to manage their symptoms with varying degrees of success. Common self-management strategies included rest, pacing of activity, heat, pelvic floor exercises (kegels), pessaries and seeking information.

"*I like to, like lay. . .put you know, hips on a pillow and lay like that, for a bit. That helps a lot, or like legs on the wall.*" (T1, P5, L31)

"*Doing the kegels correctly helps dramatically like, if I have something that's happened, like if I've been lifting a lot or doing something or not doing the exercises and I feel worse,*

*if I do the exercises regularly after three days of that I know I can feel a huge difference". (T6, P10, L11)*

*(iv) Sleep*. Most participants reported poor sleep for a variety of reasons including impact of their symptoms, they had a child who woke frequently during the night or that they were generally poor sleepers. Only three participants reported sleeping well.

"*Since I've been pregnant the first time, I've had this problem where I will wake up like at some stage in the early morning/late night and just can't go back to bed.*" *(T1, P4, L21)*

Some women linked bothersome symptoms to their poor sleep;

"*Em, sleep impacts the symptoms. Definitely. Just a lot more pain, a lot more pain and the bulging is a lot worse*" *(T6, P5, L32).*

Others noted that overwhelming tiredness impaired their ability to recognise the problem and seek help.

"*I think if baby was a better sleeper, I would have been quicker, you know. . . .I wouldn't have said I was depressed or anything. I was just chronically sleep deprived, you know, that's how I was.*" *(T7, P5, L2)*

*(v) Hormones*. Fluctuating hormones related to various times during menstrual cycles were identified as an aggravating factor for several women when they tended to be more symptomatic (pain, discomfort, heaviness dragging etc).

"*Your symptoms are much worse when you have your period. So, time of the month is big, has a big impact on it, in that it would be a lot worse.*" *(T4, P4, L54)*

*(vi) Birth mode*. The women had a high incidence of operative vaginal birth, many feeling this had contributed to, or caused their POP;

"*I had a forceps delivery and I believe that caused my prolapses*". *(T13, P2, L8)*

"*After the first birth. . .she was forceps and I had like a two-hour pushing phase, and I was having some issues at that stage*" *(T1, P1, L21)*

"*I ended up having an episiotomy and then I had forceps and I'd three or four pulls, they did three pulls of the forceps to get her out.*" *(T4, P12, L12)*

## Psychological theme

This theme consists of six subthemes: (i) knowledge (ii) beliefs & expectations (iii) anxiety/worry/fear (iv) hypervigilance (v) fear avoidance (vi) resilience

*(i) Knowledge*. Many of the women had never heard of POP before being diagnosed with it, or believed it was a condition suffered by elderly women.

"*You kinda just think people, it's older women it happens to, like I'd never have thought that my bowel would be bulging into my vaginal wall. Like. . . I didn't know that was even possible*" *(T3, P17, L15)*

Others knew about POP but unaware of the symptoms.

*"I didn't know what it would it feel like, or I didn't know what it would entail, or what the recovery would be like." (T5, P13, L19)*

Some women who had forceps births described being unaware that forceps could contribute to developing prolapse.

*"I had no idea that forceps could cause any damage." (T1, P13, L20).*

Women also reported not being aware of services available to help them with their POP or post-partum problems.

*"I wasn't aware about the HSE, that I could be referred back into the HSE or anything at that point". (T10, P4, L22)*

*(ii) Beliefs & expectations.* Most women reported seeking care for POP symptoms because they expected to be feeling better after the birth of their baby.

*"I started to notice that I wasn't feeling what I was expecting to feel after." (T2, P1, L14)*

Conversely, others described not seeking care sooner because of a prevailing belief among their family and friends that their body would never return to normal after childbirth.

*"I thought that was normal, that's what happens after baby, stitches, whatever you know". (T12, P3, L34)*

For many women, diagnosis was accompanied by warnings that their POP would likely worsen over time and that they should restrict certain activities.

*"She just basically told me not to lift anything heavy or whatever".* (T5P1L21)

*"My consultant's view was that things are only going to get worse, through menopause".* (T3P2L10)

As a result, most women expected that their prolapse symptoms would get worse over time and several expressed concerns about exacerbation of symptoms, and whether they would need surgery, especially as they approached menopause.

*"It's a very bleak picture, isn't it? You know. Like there's no, there's no cure, you know. You can't get better, it's permanent damage, you can have this surgery if you like, but you know, you could be worse off." (T14, P2, L37)*

*"It was a bit unnerving as well because you're thinking, things are going to get worse, through menopause. How bad is it going to get? Am I going to be fit for surgery?" (T3, P4, L10)*

The diagnosis of POP significantly negatively affected women's beliefs about themselves and their bodies. They reported feeling weak, broken, less feminine, less care-free and less confident.

*"You feel broken, you feel completely banjaxed, you feel like you're an old woman before your time. . . you know, you don't feel attractive." (T14, P25, L26)*

*"I definitely feel less. . .feminine, I feel. . .I feel weak, because I feel like there's something holding me back, em. . .I don't feel like the person I was before." (T3, P17, L1)*

Some of the beliefs participants held were inaccurate and unhelpful:

*"Your hips are out, your pelvis is out, the muscles are all, like you're holding on too much and your muscles are weak." (T5, P14, L33)*

*"Really the only chance of being able to go back a grade is, if you are still postpartum. . ..my understanding is I'm this far along, I will be able to manage my symptoms, but I won't ever actually like improve the prolapse itself." (T1, P11, L23)*

*(iii) Anxiety/worry/fear.* Women's unhelpful beliefs in many cases led to fear and worry. Most said that their POP and worry about it becoming worse was a major factor in deciding whether to have more children.

*"I just couldn't face the thoughts of another vaginal birth, like I was absolutely tormented with worry about being worse, ending up worse afterwards." (T14, P4, L35)*

All participants reported some degree of fear of participating in day to day or sports activities.

*"It just makes me so anxious when I feel like getting worse, like if I do something. . . like last the last few nights my toddler's teething. . ..now I'm just so anxious thinking. . ..I'm not going to be able to cope because I lift him and I end up with pain." (T6, P5, L11)*

*"I used to love doing dance and it probably would make me feel a little more anxious about kind of going back something like that." (T11, P9, L13)*

*"I do have the fear that if I push myself too much I'll make it worse." (T1, P11, L15)*

*(iv) Hypervigilance.* Hypervigilance and thinking frequently about their symptoms were common experiences. Women described their diagnosis as something they had to carry, something that was always there and something they had to consider before partaking in activities, especially exercise.

*"Subconsciously you think of it all the time, like so you almost think like, what can I do to fix it like, you know, all the time." (T14, P3, L13)*

*"If I do, a particularly heavy like, pilates session or something, I'm noticing it the next day and it's really interesting right because I don't know if I'm noticing it because I'm worried about it, or I'm noticing it. . . because I can feel it." (T2, P10, L34)*

*"I'd say I think about something related to my prolapse three or four times an hour." (T5, P8, L34)*

*(v) Fear avoidance.* All the women described avoidance of certain activities either due to exacerbation of symptoms or fear of worsening their symptoms long term. The women described reducing or avoiding jobs around the house and avoiding lifting, carrying, or engaging in certain play activities with their children.

*"To be honest, even some normal stuff like hanging stuff on the line, I have something beside the clothesline, so that I don't have to bend to the ground, like things like that, that I do*

*actually have to think quite a bit. Because like, if I think about it, I don't have as much pain, if I don't think about it, I end up with more pain." (T6, P6, L33)*

*"Going on the trampoline with my kids, definitely not. . . .I've worried about doing more damage to my prolapse in moving something to where it shouldn't be. . .so, running and I suppose, in terms of our sex life, having intercourse. I'm conscious. . .I'm nervous of the impact that could do in making things worse." (T3, P15, L19)*

They also reported curtailing or avoiding many types of exercise.

*"I can't go for a run; I can't do these things I would do to kind of relieve stress as well. You know. . . that kind of physical stress relief. I feel like I don't really have that." (T2, P4, L27)*

*"No, no exercise like I literally do nothing, which is desperate, but it's very hard to find an exercise that is prolapse friendly." (T4, P6, L2)*

Women expressed difficulty performing duties at work or feeling that they could not return to work in a role they had previously occupied.

*"When I can't empty my bowel properly and I need a little step to put my feet on or you know, the flatulence from the, all the different cocktail of laxatives, I'm taking is really bad, I think how the hell, can I be in any kind of a workplace the way I am" (T14, P16, L25)*

*(vi) Resilience.* Despite the challenges of living with POP, the women demonstrated remarkable resilience, expressing acceptance, coping strategies for managing their POP and taking ownership of their condition through information and support from their healthcare professionals and community.

*"You have to take it on yourself, you know, and I think with pelvic floor, like, you have to do a bit of work." (T7, P16, L24)*

*"I kind of have to shake it off and go, no you're fine and I might think of it a couple more times and I'm like, no, no, it's fine, it's grand, you're okay, you can keep going. Because. . . 21 months on, I do feel stronger, and I am feeling optimistic." (T2, P17, L23)*

*"I kind of feel like that's the whole thing for me, is that, you know, information is power, I was able to go off and do my own research from there." (T3, P6, L44)*

## Social theme

This theme encompasses seven subthemes: (i) role as a mother (ii) role in the home (iii) intimacy/sexual life (iv) the stigma of POP (v) participation in sport and exercise (vi) work life & relationships.

*(i) Role as a mother.* Difficulties in caring for their children was a recurring problem for most women. They reported being unable to or worried about lifting or carrying their children, avoiding engaging in certain types of play, difficulties fitting recommended rehabilitation around their family life and having less patience with their children due to discomfort or being mentally preoccupied with their symptoms.

*"I was very conscious of not being able to lift and not being able to care for him in the same way that I would have wanted to, because I wasn't able to run round and just. . .lift him up on things, lift him up for a hug and stuff like that." (T3, P9, L33)*

*"Trying to manage time, with a small baby, you know, trying to do. . .. that electrocuting your vagina for half an hour every day." (Joanne, T7, P3, L34)*

*(ii) Role in the home.* Managing household chores was also difficult for many women. In some cases, they had been advised by a HCP to avoid the activity and in others they avoided it because of fear of causing or increasing symptoms. They reported difficulties mainly with lifting, pulling and pushing (ie. carrying shopping bags, emptying bins, vacuuming and moving small furniture).

*"There's loads of things I can't do around the house, I can't empty the kitchen bin because . . .it's too heavy. I'm not supposed to carry the hoover upstairs but I do and then I pay for it the next day. Even just doing like, more than an hour of housework, I'd be in bits the next day." (T5, P6, L6)*

*"I'm quite conscious of the fact of hoovering that you're, you're pushing or pulling. . . I can't do the big shop, shall we say, because I can't push a trolley and I'm not meant to lift more than 10 pounds now post-surgery." (T4, P1, L13)*

*(iii) Intimacy/sexual life.* Most women expressed gratitude that their partner was understanding of POP, the symptoms and how it affected daily life and their intimate relationship, although they also described conflict in their relationship due to preoccupation or frustration with their POP symptoms.

*"I think he was very understanding. . . just take whatever time you need and. . . we'll take it as slow as you want to whenever you want, and he was very understanding that way." (T9, P8, L36)*

*"I think you know, we talked about that kind of sharpness that you get. That would probably come out more to him. More often than not that comes out to him more so than the kids and. . . he understands why but it's still not not great for him." (T2, P6, L3)*

They also reported avoiding sexual activity because of pain or fear of pain, low libido because of self-consciousness about their body or needing a health professional to confirm it was safe.

*"I don't want to say to my husband, no actually I don't want to have sex, because I can feel stool there, in my vagina and. . . I don't want to say that, you know?" (T14, P14, L28)*

*"I wasn't able to have sex, penetration was just impossible." (T11, P4, 23)*

*'We've held off on doing anything until I can see a specialist because I feel very nervous about it so yeah, that's affecting that but he's, like I said very, very understanding." (T1, P6, L5)*

*(iv) The stigma of POP.* All the women highlighted a culture of silence and shame around POP and other pelvic health and sexual issues. They felt that though POP is common, awareness of the condition is low because of the stigma attached to these conditions.

*"I feel like people 'don't talk about it enough. . ..but I'm like yeah but if we don't talk about it, then nobody else is going to ever know about it, so . . .it's a bit like miscarriage, it's a bit like all these other things that we don't talk about, it then makes it harder for people to know where to get the right information and to ask the questions." (T2, P21, L23)*

"*But I've never heard any other woman in my life talk about those things.*" (T10, P13, L8) "*It just seems like there's a sense of shame, around this stuff in this country.*" (T12, P11, L16)

Conversely, many of the women expressed frustration at what they perceive to be the normalisation of pelvic health issues, among women themselves, HCPs and especially popular media.

"*I hate that people think that it's normal for this to happen to your body afterwards. And like if you're my age (. . . in your mid 30s), I would have to live with that for the rest of my life, just because somebody thinks it's normal. I think that's completely unacceptable.*" (T2, P15, L3)

"*If women say that people make a joke of it. There's ads on television telling you, 'don't worry about it, just buy these black nappies and you'll be fine. You know?*" (T14, P7, L5)

*(v) Societal attitudes to women and women's bodies*. The pressure to conform to an idealised perception of new motherhood and recover quickly after the birth of their babies was a common observation among the women.

"*I think you kind of have to. . . .try and do as much as I can, prove I can cope with this new baby.*" (T3, P15, L40)

"*You feel like you need to do everything. . . have to get your body back, . . .your house back and you have to. . .breastfeed beautifully and. . .have your hair washed going out the door.*" (T3, P16, L12)

"*In work we'd have women who had their baby and then they're like in creche collecting the toddler two days later. . . it shouldn't be the norm. . .people do it because they have no choice. . . the idea is that. . .you just pop it out and go back. . . lose your weight and go back to normal.*" (T6, P11, L11)

Many remarked on societal attitudes to women and their bodies that affect healthcare for women.

"*There's very little appreciation, I feel for medical conditions and issues that women have.*" (T3, P8, L5)'

"*And isn't it funny, like there's probably plenty of men with incontinence but it's always a woman in the ad, you'd never see a man like happy out, running, playing soccer in a nappy.*" (T8, P18, L33)

"*Women are oppressed. I don't know, probably the same reason why they have to use blue ink on a on a sanitary pad ad. Women's bodies are not there to function.*" (T7, P12, L3)

They also perceived the healthcare system as being male dominated and that women's bodily integrity is seen as being secondary to the wellbeing of their babies.

"*This idea, as long as the baby arrives healthy. . .*" (T7, P17, L30)

"*I think, you know, it comes back to (and I know it's a very generalistic view) but it comes back to my view those it's a male dominated profession. And, there's just. . .there's just not the consideration. . .eh, given to women particularly.*" (T3, P17, L35)

*(vi) Work.* Approximately half the women interviewed were currently working outside of the home. None reported leaving their jobs because of their POP symptoms, although they described associated physical difficulties, such as symptoms being aggravated by occupational lifting, walking to and from work or long hours on their feet.

*"I've a commute. . ..and that means carrying my laptop bag and the walk is maybe 10, 15 minutes both ways so that would. . . that would affect me." (T4, P1, L57)*

Others reported that their symptoms made it difficult for them to feel they were fulfilling their role to the best of their ability at work.

*"I have a really, really strong work ethic. . ..I really, really find it hard to ask for help so when I couldn't do my job, or these extra elements to my job, I felt that I wasn't doing my best." (T3, P10, L41)*

The women who were not working outside of the home were ambivalent about how they might manage if they decided to return to the workplace.

*"I wonder would I have worn heels every day, I wonder, would have been able to do what I would have done before. Like would I be worried about getting tired?" (T2, P6, L44)*

*"I was thinking or applying for a job in a nursing home nearby. But actually. . .I wouldn't be able to lift the patients. Because you're not meant to lift anything heavy. And you're not meant to be on your feet for too long either. . .. So, I was thinking, I just decided not to apply for the job then." (T5, P9, L19)*

*(vii) Participation in sport and exercise.* For many, sport and exercise, particularly running, was an important part of their lives and often used as an outlet for stress. Having POP meant that for some, increased symptoms with increased PA, and for others, the advice of HCPs meant they were forced to stop or significantly curtail their usual PA.

*"I love going swimming. . . but I'd imagine that at some point, I was going to get feel self-conscious about it or think that it was getting worse from it and so. . .why start if you're going to have to stop?" (T11, P9,'L18)*

*"I don't like going on hikes or anything, unless I am very well prepared with taking a little portable toilet and or she-wee so that I can pee standing up." (T10, P10, 15)*

Women reported feeling 'weak', not attaining their pre-pregnancy fitness levels because of worry about exacerbating their POP, not having an outlet for their stress, and missing their sport.

*"I was very active and . . . went to the gym five days a week, etc, and plus like I said, I am still walking a lot but I'm. . . like with body pump I was doing five kilo dumbbells so ten kilos at a time which isn't CrossFit levels or anything, but I even am afraid to do that much now." (T1, P11, L3)*

*"I do miss that amount of exercise, I miss the euphorias and the dopamines and all that, that it gives you." (T9, P1, L55)*

Several women cited difficulty finding 'prolapse-friendly' exercise or conflicting advice from HCP as barriers to continuing or recommencing exercise.

"*I wouldn't dare go for a hike or anything like that, em,'just wouldn't, wouldn't do it, wouldn't run. No, no exercise like I literally do nothing, which is desperate, but it's very hard to find an exercise that is prolapse friendly*" (T4, P6, L2)

"*I've had contradicting recommendations, em on whether rowing is good or bad like, so. . .*" (T9, P11, L36)

## Discussion

This study explored younger pre-menopausal women's experiences of living with POP. Three themes with associated subthemes emerged; biological (physical symptoms, aggravating factors, self-management, sleep, hormones and birth mode), psychological (knowledge, beliefs and expectations, anxiety, fear and worry, hypervigilance, fear avoidance and resilience) and social impact (role as a mother, role in the home, intimacy/sexual life, the stigma of POP, societal attitudes to women and their bodies, work, participation in sport & exercise). Many of these main themes and subthemes reflect those reported in research on other chronic conditions such as low back pain [27–29], fibromyalgia [30–32] and vulvodynia [33–35].

Some women reported guilt and sadness that they were unable to mother as they would have liked, many described attempting to avoid lifting, carrying and some play and physical activities with their children. This avoidance of perceived 'harmful' activities has previously been described in the literature [11, 12, 36–38].

Others described mental preoccupation with their symptoms and having less patience with their children as a result. However this is not consistently borne out in the literature; whilst some of the limited qualitative literature focusing on pre-menopausal women concurs [11, 39], the results of the current study contrast with Ghetti et al's findings [14], where some participants described primarily physical symptom bother and little to no emotions related to prolapse. Ghetti's population had an average age of 60±10 years, which differs to the demographic of the current study (age range 32–41, mean 37 years) and that of Mirskaya et al [11] (age range of 25–45, mean 27 years). Ramage's [39] sample also had a younger mean age at 53 years and reported similar findings to those of the current study.

It is accepted that risk of developing POP increases with age [40]; however, there has been little enquiry as to how age affects POP symptom bother with POP research focusing either on postmenopausal women with POP or including samples of women of widely varying age. One small study [41] examined the relationship between age and POP bother, reporting that women in their 50s and 60s are at higher risk for bother as a result of POP; however, the mean age of women in this study was 60±11 years. Another large study of nulliparous women [42] found increased bother in 55 to 64 year olds compared to younger age groups. Given the fact that the participants in this study were not mothers, and the findings of this and other work [11, 37, 39] that the mothering role is significantly impacted by symptoms of POP, further research is needed to examine the impact of POP among younger women with children.

Similar to the findings of Mirskaya [11] and Toye [37], women in the current study reported that their POP diagnosis was a consideration when planning further pregnancies and feared the physical demands of mothering could contribute to worsening their POP.

Fear was an emotion described by most women. It appeared to be compounded by lack of knowledge about POP and negative and limiting expectations of both women and their HCPs

regarding POP prognosis. Also significant were the limitations placed on women by themselves in addition to HCPs regarding restricting physically demanding activities, in particular their participation in exercise and sport. This type of adaptive behavioural strategy in women with POP has been previously described [11, 12, 36, 38].

Research in other chronic conditions has shown that fear avoidance beliefs and catastrophisation are often more disabling than the condition itself [43]. With regard to POP, associations between higher levels of catastrophizing and higher levels of bother in women with POP have been reported [44].

The CSI scores, indicating some level of CS in 12 of the 14 women (86%) may explain the high level of distress among this sample. Previous research has reported lower rates of CS in women with POP (40%) [8]; mean age was 60 years compared with 37 years in the current study. Poor sleep was reported by many of the study participants, due to the fact that they were mothers of young children. Sleep disturbance has been associated with CS in research on irritable bowel syndrome [45], migraine [46] and fibromyalgia [47], which were co-morbidities for some of the participants. It has been demonstrated that women with CSI are less likely to have favourable outcomes after POP surgery [48] and research in chronic pain patients with CSI has shown that treatment strategies aimed at targeting local structures are typically of little value in those with predominant CS bother [49]. This research suggests a more "central" or psychosocially-informed approach (stress management, sleep management, graded activity/graded exercise therapy, graded exposure and a recognition of the biopsychosocial perspective) targeting brain and top-down mechanisms seems warranted for treatment of CS. Therefore evaluation of CS to inform assessment, diagnosis and treatment of POP may help to improve outcomes in this cohort.

The POP literature has noted a lack of data regarding POP progression and regression, as well as widespread acceptance that POP is a progressive condition, which has guided its management to date [50]. However, research has demonstrated that POP regression occurs in 19–48% of women with stage 1 or 2 prolapse without intervention over three to eight years [5], while other research demonstrated 21% regression with the use of vaginal support pessaries [51] and both subjective and objective POP improvement with pelvic floor muscle training (PFMT) [52].

In other common chronic conditions such as low back pain it has been reported that HCPs have the strongest influence on patient's beliefs about their condition [53]. There is the potential that fear avoidance beliefs among HCPs regarding POP may have a negative impact on patient coping and self-management, leading to poor outcomes and persistent disability as observed in low back pain patients with strong fear avoidance beliefs [54].

More strenuous physical activity (PA) was identified for many of the women as a trigger for their POP symptoms, and most reported either reducing or completely avoiding it as a consequence. There is a lack of evidence to support the assertion that moderate or high intensity PA is associated with increased incidence of POP [55, 56]. Some evidence suggests that PA may have a protective effect for some types of pelvic dysfunction in that exercising women tend to have similar or stronger pelvic floor muscles than non-exercising women [57]. By completely avoiding all strenuous PA, women with PD may inadvertently be further negatively impacting their pelvic floor function in the long term. Women with PD have been shown to discontinue or modify PA participation as a result of pelvic floor symptoms, (urinary incontinence, urinary urgency or loss of bowel control and less often due to POP symptoms (vaginal bulging or heaviness) [58].

The increase in sedentary behaviour in women with POP has the potential to be detrimental to the individual and healthcare system in the short term, increasing the risk of health-related complications such as gestational diabetes, pregnancy-induced hypertension, postpartum

depression, urinary incontinence and preeclampsia in future pregnancies [59–62]. Similar to the management of other chronic conditions women with POP should be supported and encouraged to actively self-manage their condition through early access to specialist treatment to address accompanying pelvic floor dysfunction, education, pacing and encouraging PA (tailored to the individual patient's ability and circumstances) throughout all life stages.

In addition to the negative impact on women themselves, it is accepted that children's activity levels may be influenced by their parents modelling behaviour, particularly that of mothers, with girls of mothers who participate regularly in PA engaged in more sports and practicing PA more times per week [63–65].

Fear and self-consciousness about their bodies also affected women's sexual lives and intimate relationships, as has been reported elsewhere [66]. Research suggests that these issues are likely a product of societal attitudes to women's bodies rather than the anatomic features of POP [67] and that sexual function remains unchanged for many women following POP surgery [68, 69]. Other research demonstrates an association between increased PA and improved sexual function in peri- and post-menopausal women [70], which warrants further exploration as a cost effective, low risk intervention, particularly for women with mild to moderate POP.

Increased education, knowledge and awareness about POP among women of all ages, particularly prior to becoming pregnant has been highlighted by women with POP as a key area for improvement [11, 16, 71, 72] and has been shown to be effective in reducing fear avoidance and increasing function in people with chronic health issues [73–77].

Women have also highlighted healthcare system issues such as difficulty accessing reliable information about POP, inadequate HCP knowledge about POP and issues with access to specialised care for POP due to cost or lack of services [16, 71]. A general lack of awareness of POP, even among women at risk of developing the condition has also been noted [72]. Little research has focused on the natural history of POP or the effects of exercise on pelvic organ support [50] in women with and without POP and there are no guidelines for exercise in women with pelvic dysfunction for HCPs to refer to.

Women in the current study viewed obstetrics and gynaecology as a male dominated field of medicine and questioned how male gynaecologists could identify with or understand the impact of POP symptoms on women's lives. This reflects the findings of research regarding a higher patient satisfaction and preference for female-led obstetric or gynaecological care [78–81] and is contrary to other studies where the majority of women had no preference for HCP gender [82–84]. Better communication between HCPs and shared decision making has been shown to improve patient satisfaction and health outcomes [85] and to be a priority for patients [86] and should be promoted among all HCPs to improve patient self-efficacy and health outcomes.

Given the uncertainty regarding the aetiology of POP, investigation of strategies to prevent its development has been limited. Pelvic floor muscle training (PFMT) [87, 88], use of local oestrogen treatments [89], elective caesarean in women at higher risk of POP, minimizing forceps deliveries and episiotomies and allowing passive descent in the second stage of labour [89–91] have been suggested as possible strategies for secondary prevention of POP.

Other research has focused on the use of a scoring system based on PD risk to allow informed choice for women regarding PD prevention during pregnancy and labour, which was found in a small sample of women and HCPs to be feasible and acceptable [92]. Further research is needed to investigate the effectiveness of conservative and lifestyle measures, consideration of future PD and increased utilization of shared decision-making when discussing birth interventions with women and surgical interventions for the prevention and treatment of POP.

Healthcare systems, along with all HCPs, and education systems should promote awareness of normal pelvic floor function and where to seek help if required. This could be achieved by increasing awareness of pelvic health and the impact of POP among both male and female HCPs, incorporating information into sexual health programmes in schools, screening at GP and other health visits and public awareness campaigns [93–96]. More knowledge among women regarding symptoms of POP and access to evidence-based and accessible information could also help to reduce catastrophizing and distress [97]. Ensuring adequate levels of health literacy would be important to ensure self-management in their own health.

Clinical decision-making pathways could assist GPs, public health nurses and other HCPs in directing patients to appropriate services where needed. Increased knowledge and confidence in the area through education is also needed for HCPs, as research has shown that primary care physicians may lack knowledge or feel less comfortable screening for or treating POP than other pelvic health disorders [98, 99]. Increased and early access to evidence-based conservative treatment such as PFMT and pessary fitting is also important given the significant benefits associated with both in the research available to date [100, 101].

## Limitations

The sample were recruited from an online social media peer support group, and by their membership had actively sought further information and support with their condition. This may imply a level of self-efficacy, health literacy and familiarity with and access to technology, not necessarily reflective of all women with POP.

It is possible also that women with more bothersome POP, regardless of objective POP stage, were more likely to enroll in the study with the expectation of being able to air their distress. With regard to CSI scores, the sample size of the current study is too small to draw firm conclusions regarding the impact of CS on symptom distress in this population.

The very high levels of distress and high CSI scores may be influenced by the fact that this research was carried out during the Covid 19 pandemic.

## Conclusion

The results of this study show that POP has a significant biopsychosocial impact on women, in particular impacting their perceptions of their ability to parent and to participate in PA. High levels of distress among women and low levels of knowledge suggest a psychologically informed approach to treatment may be beneficial for this cohort. Increased investment in education and awareness of POP both among the public and HCPs to adequately prevent and treat this common condition is needed.

## Author Contributions

**Conceptualization:** Louise Carroll, Cliona O' Sullivan, Catherine Doody, Carla Perrotta, Brona Fullen.

**Data curation:** Louise Carroll.

**Formal analysis:** Louise Carroll, Brona Fullen.

**Funding acquisition:** Brona Fullen.

**Investigation:** Louise Carroll, Cliona O' Sullivan.

**Methodology:** Louise Carroll, Cliona O' Sullivan, Catherine Doody, Carla Perrotta, Brona Fullen.

**Project administration:** Louise Carroll.

**Resources:** Brona Fullen.

**Software:** Louise Carroll.

**Supervision:** Cliona O' Sullivan, Catherine Doody, Carla Perrotta, Brona Fullen.

**Validation:** Louise Carroll, Brona Fullen.

**Visualization:** Louise Carroll, Cliona O' Sullivan, Catherine Doody, Brona Fullen.

**Writing – original draft:** Louise Carroll.

**Writing – review & editing:** Louise Carroll, Cliona O' Sullivan, Catherine Doody, Carla Perrotta, Brona Fullen.

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
