## [Decision Letter · Decision Letter 0]

23 Aug 2022

PONE-D-22-22443Pelvic Organ Prolapse: The Lived ExperiencePLOS ONE

Dear Dr. Carroll,

Thank you for submitting your manuscript to PLOS ONE. After careful consideration, we feel that it has merit but does not fully meet PLOS ONE’s publication criteria as it currently stands. Therefore, we invite you to submit a revised version of the manuscript that addresses the points raised during the review process.

We look forward to receiving your revised manuscript.

Kind regards,

Antonio Simone Laganà, M.D., Ph.D.

Academic Editor

PLOS ONE

Journal Requirements:

Additional Editor Comments:

The reviewers have expressed positive comments regarding your article, raising only few concerns. Considering this point, I invite authors to perform the required minor revisions.

Reviewers' comments:

Reviewer's Responses to Questions

**Comments to the Author**

1. Is the manuscript technically sound, and do the data support the conclusions?

Reviewer #1: Yes

Reviewer #2: Yes

2. Has the statistical analysis been performed appropriately and rigorously? 

Reviewer #1: No

Reviewer #2: N/A

3. Have the authors made all data underlying the findings in their manuscript fully available?

Reviewer #1: Yes

Reviewer #2: Yes

4. Is the manuscript presented in an intelligible fashion and written in standard English?

Reviewer #1: No

Reviewer #2: Yes

5. Review Comments to the Author

Reviewer #1: I read with great interest the Manuscript titled “Pelvic Organ Prolapse: The Lived Experience” (PONE-D-22-22443), which falls within the aim of this Journal.

In my honest opinion, the topic is interesting enough to attract the readers’ attention. Nevertheless, authors should clarify some point and improve the discussion citing relevant and novel key articles about the topic.

Authors should consider the following recommendations:

- Manuscript should be further revised by a native English speaker

- What are the actual clinical implications of this study? it is important to report the results obtained by the authors in the context of clinical practice and to adequately highlight what contribution this study adds to the literature already existing on the topic and to future study perspectives

- Does this manuscript conform the Enhancing the QUAlity and Transparency Of health Research (EQUATOR) network guidelines? It would be mandatory to declare about this element.

- Was this study registered? I could not find any information about this point.

- Among POP, cystocele represents of the most challenging condition: the prolapse of anterior compartment, indeed, could be treated by both prosthetic surgery and native tissue repair. I suggest to discuss pro and cons of these two approaches, referring to: PMID: 33813235; PMID: 26801794.

- I would invite authors to discuss the outcomes on bladder and urinary functions after advanced laparoscopic surgery, such as for instance pelvic exenteration (PMID: 26807639) and laterally extended endopelvic resection (PMID: 32332122) for gynecological malignancies

Reviewer #2: Feedback on PLOS ONE manuscript # PONE-D-22-22443

1. The manuscript appears technically sound but the authors should provide evidence that they have used a checklist for qualitative research such as the COREO or SRQR.

2. qualitative study thus no statistical analysis applicable.

3. -

4. The authors should address the following:

- UCD should be written in full on the title page

-Line 36: in the summary you state that “Most had Grade 1-2 POP”. .However, 3/14 ie 21.4% had stage 3 POP so I think that this statement should be clear that POP stages were 1-3.

-line 101: what preventative strategies do you propose for POP given that vaginal birth is the greatest risk factor..You might mention your thoughts on this in the discussion

-line 104 - the word 'summary' is not needed

-PI needs to be defined/written out in full in line 106 where I think it is first mentioned and not first in line 119.

-the authors have used the term 'grade' to describe the severity of prolapse instead of 'stage'. POPQ uses the term ‘staging’, not grading. This needs to be corrected in a number of places throughout the manuscript.

-line 155: insert the word 'to' (participants to elaborate)

-line 160: you conducted the interviews during Covid lockdowns – this may have increased the perceived distress of the women and should be included in the possible explanations for the high CSI scores and high distress.

- Table 2 has a heading for POP type /compartment but the dominant compartment is not reported for each subject. Please include this information for the reader.

- You have not reported any other pelvic floor dysfunctions e.g urinary incontinence, faecal incontinence, pelvic pain, which may be contributing to the women’s distress. Please include these details and discuss this in light of the fact that the subject report of distress was high although severity was low. There needs to be some exploration of this apparent contradiction.

- the CSI is reported in Table 2 and in lines 204-5 in the results but not in the discussion although the scores are astonishingly high. The presence of CSS is a relatively new aspect of POP assessment -well done for including it - but you should address your findings in some detail and discuss/interpret them. Do the high CSI scores help explain the high levels of distress despite low severity? Lack of sleep?

- There are 4 tables which are not numbered correctly. The labels would be better placed above the table not below.

-line 208: I can count only three women where general health perception is not reported, not four. Please check.

- there are numerous typographical errors in the transcriptions which the authors need to correct.

- subjects are reported as using pessaries as well as doing PFMT - it would be interesting for the reader to know which subjects used a pessary. Please add to Table 2. Are those using a pessary less distressed perhaps?

-line 555: please refer to the recent IUC document on POP aetiology regarding the relationship of age with POP

- line 575: …as well as widespread acceptance that POP is a progressive condition, which has guided its management to date42.

-line 576: does POP not increase with age so that the women’s fears of progression are justified?

-line 582: please check that you have spelt out PA previously..

-line 586: I do not think that this assertion is correct that…No association has been found between moderate or high intensity PA and increased incidence of POP45-47

It suggests that this has been thoroughly investigated and that there is no association rather than that there is a lack of evidence to be able to draw firm conclusions.

Firstly ref 45, Bo et al 2020 state “The few studies available to assess the association between exercise and POP .. are inconsistent in their conclusions.”

Secondly ref 46 Forner et al showed that women lifting heavier weights did not have more POP. However, this does not mean that lifting heavier weights is not a potential problem for women with POP. The women performing lower impact/weight exercise in this study had self-selected this level of activity because increasing weights made their POP symptoms worse.

Thirdly ref 47 is in a population of women with SUI and I don’t think these findings can be extrapolated to women with POP. In this study, continent women did more PA while incontinent women did less PA, possibly precisely because the exercise made them leak urine. I think that this reference should be deleted and your statement corrected to reflect the actual state of the evidence.

- line 591: Similar to the management of other chronic conditions women should be supported and encouraged to actively self-manage their condition through education, pacing and encouraging PA throughout all life stages. You seem to be supporting the view that women with symptomatic POP should be encouraged to engage in PA but without considering the fact that PA makes them more symptomatic and that it is impossible to say from the current exercise what type of level of exercise can be recommended for women with different types and degrees of POP. I think that this needs to be clarified. So it is really like other chronic conditions where exercise is beneficial and won’t lead to an exacerbation of the condition or symptoms?

- line 628 - the grammar needs to be reviewed and corrected

-line 639: The study sample were recruited –gr: the sample was recruited

- line 639- Limitations: I think that a further limitation of the study and a potential selection bias, is that women with more bothersome POP, regardless of severity may have responded to the call, with the expectation of being able to vent their distress. This may also account for the higher bother and CSI scores?

- reference list needs to be proof read as some titles do not adhere to Vancouver style: e.g # 37,38,61,62,67

6. PLOS authors have the option to publish the peer review history of their article (what does this mean?). If published, this will include your full peer review and any attached files.

Reviewer #1: No

Reviewer #2: No

---

## [Author Response · Author response to Decision Letter 0]

13 Sep 2022

Dear Reviewers, 

Thank you for your feedback. 

I would like to address your points raised on reviewing our manuscript “Pelvic Organ Prolapse: The Lived Experience” (PONE-D-22-22443). 

Reviewer’s Comments

I read with great interest the Manuscript titled “Pelvic Organ Prolapse: The Lived Experience” (PONE-D-22-22443), which falls within the aim of this Journal.

In my honest opinion, the topic is interesting enough to attract the readers’ attention. Nevertheless, authors should clarify some point and improve the discussion citing relevant and novel key articles about the topic. Authors should consider the following recommendations:

1. Manuscript should be further revised by a native English speaker

The manuscript has been thoroughly revised and reviewed by two native English speakers.

2. What are the actual clinical implications of this study? it is important to report the results obtained by the authors in the context of clinical practice and to adequately highlight what contribution this study adds to the literature already existing on the topic and to future study perspectives

On revision the authors have discussed evaluating possible CS (Line 586), a ‘central’ or psychosocially informed approach (Line 582), early access to specialist treatment to address accompanying pelvic floor dysfunction (Line 627), education, pacing and tailoring general physical activity to the individual (Line 516). We have also added information on the current evidence base regarding conservative treatment and the importance of early access to same (Line 647)

3. Does this manuscript conform the Enhancing the QUAlity and Transparency Of health Research (EQUATOR) network guidelines? It would be mandatory to declare about this element.

Yes the manuscript conforms to the SRQR guideline. This information has been added (Line 99).

4. Was this study registered? I could not find any information about this point.

The study was not registered.

5. Among POP, cystocele represents of the most challenging condition: the prolapse of anterior compartment, indeed, could be treated by both prosthetic surgery and native tissue repair. I suggest to discuss pro and cons of these two approaches, referring to: PMID: 33813235; PMID: 26801794.

I would invite authors to discuss the outcomes on bladder and urinary functions after advanced laparoscopic surgery, such as for instance pelvic exenteration (PMID: 26807639) and laterally extended endopelvic resection (PMID: 32332122) for gynecological malignancies

The authors have reviewed your recommendation regarding the merits of conservative or surgical interventions. We recognise the significant benefits of surgery for selected women with POP. However, as the focus of the manuscript is on the lived experience of women with POP we do not believe that this lies within the scope or purpose of this manuscript. 

Reviewer 2

1.The manuscript appears technically sound but the authors should provide evidence that they have used a checklist for qualitative research such as the COREO or SRQR.

The SRQR was used and reference to the SRQR has been added in the manuscript (Line 99)

Title (Line 2)

Abstract (Line 20)

Problem Formulation (Line 61-90)

Purpose/Research Question (Line 96)

Qualitative Approach & Research Paradigm (Line 100, 138)

Researcher Characteristics & Reflexivity (Line 151)

Context (Line 155)

Sampling Strategy (Line 122)

Ethical (Line 104)

Data Collection (Line 120)

Data Analysis (Line 160)

Trustworthiness (Line 178)

Findings Synthesis & Interpretation (Line 233)

Links to empirical data (quotes used throughout results section)

Discussion integration with prior work – yes

Limitations (Line 688)

Conflicts of interests – none

22.UCD should be written in full on the title page

UCD has been written in full on the title page 

3.Line 36: in the summary you state that “Most had Grade 1-2 POP”. .However, 3/14 ie 21.4% had stage 3 POP so I think that this statement should be clear that POP stages were 1-3.

This sentence has been adjusted for clarity and now reads: ‘All had Grade 1-3 POP’ (Line 36)

4. line 101: what preventative strategies do you propose for POP given that vaginal birth is the greatest risk factor. You might mention your thoughts on this in the discussion

Current recommendations/suggestions on prevention have been added to the discussion section. (Line 647).

5.line 104 - the word 'summary' is not needed

This has been amended.

6.PI needs to be defined/written out in full in line 106 where I think it is first mentioned and not first in line 119.

This has been amended (Line 101).

7.The authors have used the term 'grade' to describe the severity of prolapse instead of 'stage'. POPQ uses the term ‘staging’, not grading. This needs to be corrected in a number of places throughout the manuscript.

This has been amended throughout the paper.

8.line 155: insert the word 'to' (participants to elaborate)

This has been amended.

9.Line 160: you conducted the interviews during Covid lockdowns – this may have increased the perceived distress of the women and should be included in the possible explanations for the high CSI scores and high distress.

A section has been added (Line 574) in the discussion regarding the CSI scores including possible explanations and implications. 

10.Table 2 has a heading for POP type /compartment but the dominant compartment is not reported for each subject. Please include this information for the reader.

Table 2 has been amended to include this information

11.You have not reported any other pelvic floor dysfunctions e.g urinary incontinence, faecal incontinence, pelvic pain, which may be contributing to the women’s distress. Please include these details and discuss this in light of the fact that the subject report of distress was high although severity was low. There needs to be some exploration of this apparent contradiction.

These details have been added to Table 2 and this point has been added to the discussion (line 618).

12. the CSI is reported in Table 2 and in lines 204-5 in the results but not in the discussion although the scores are astonishingly high. The presence of CSS is a relatively new aspect of POP assessment -well done for including it - but you should address your findings in some detail and discuss/interpret them. Do the high CSI scores help explain the high levels of distress despite low severity? Lack of sleep?

This has been added to the discussion (Line 578) and limitation (Line 679) sections.

13.There are 4 tables which are not numbered correctly. The labels would be better placed above the table not below.

All tables have been re-numbered, and labels placed above the table.

14.line 208: I can count only three women where general health perception is not reported, not four. Please check.

This has been amended (Line 204).

15. there are numerous typographical errors in the transcriptions which the authors need to correct.

The authors have reviewed the manuscript for typographical errors and corrected them.

16.Subjects are reported as using pessaries as well as doing PFMT - it would be interesting for the reader to know which subjects used a pessary. Please add to Table 2. Are those using a pessary less distressed perhaps?

This information has been added (using a star beside the participant number and adding this to the key below Table 2).

17. line 555: please refer to the recent IUC document on POP aetiology regarding the relationship of age with POP

We agree with this reviewer that the literature shows age as a factor in the development of POP. However most POP research either focuses on a broad range of women in terms of age or menopausal or postmenopausal women. We believe that POP may affect younger women differently in terms of biopsychosocial bother and this may not be captured in some of the studies because of this. We have referred to the IUC document regarding this observation.

18. line 575: …as well as widespread acceptance that POP is a progressive condition, which has guided its management to date42. -line 576: does POP not increase with age so that the women’s fears of progression are justified?

While we agree that POP may be a progressive condition for some women, there is evidence that has been included in the revision of the discussion (Line 604) that there is the potential for POP to regress over time as well as progress (with or without treatment). The authors believe that using a psychologically informed approach to treatment, this information should also be given to women instead of only communicating the assumption that their POP will progress which may contribute to feelings of hopeless and not adhering to home exercise programmes/advice given etc

19. line 582: please check that you have spelt out PA previously..

This has been amended.

20. line 586: I do not think that this assertion is correct that…No association has been found between moderate or high intensity PA and increased incidence of POP45-47

It suggests that this has been thoroughly investigated and that there is no association rather than that there is a lack of evidence to be able to draw firm conclusions.

Firstly ref 45, Bo et al 2020 state “The few studies available to assess the association between exercise and POP .. are inconsistent in their conclusions.”

Secondly ref 46 Forner et al showed that women lifting heavier weights did not have more POP. However, this does not mean that lifting heavier weights is not a potential problem for women with POP. The women performing lower impact/weight exercise in this study had self-selected this level of activity because increasing weights made their POP symptoms worse.

Thirdly ref 47 is in a population of women with SUI and I don’t think these findings can be extrapolated to women with POP. In this study, continent women did more PA while incontinent women did less PA, possibly precisely because the exercise made them leak urine. I think that this reference should be deleted and your statement corrected to reflect the actual state of the evidence. 

This section has been corrected to reflect the reviewers comments (Line 601).

21. line 591: Similar to the management of other chronic conditions women should be supported and encouraged to actively self-manage their condition through education, pacing and encouraging PA throughout all life stages. You seem to be supporting the view that women with symptomatic POP should be encouraged to engage in PA but without considering the fact that PA makes them more symptomatic and that it is impossible to say from the current exercise what type of level of exercise can be recommended for women with different types and degrees of POP. I think that this needs to be clarified. So it is really like other chronic conditions where exercise is beneficial and won’t lead to an exacerbation of the condition or symptoms?

The authors have clarified this section to recommend that PA should be recommended and tailored based on a patient’s individual circumstances (Line 613).

22. line 628 - the grammar needs to be reviewed and corrected

The grammar has been reviewed and corrected.

23.line 639: The study sample were recruited –gr: the sample was recruited

The grammar has been corrected.

24. line 639- Limitations: I think that a further limitation of the study and a potential selection bias, is that women with more bothersome POP, regardless of severity may have responded to the call, with the expectation of being able to vent their distress. This may also account for the higher bother and CSI scores?

The authors agree with this observation and have amended the limitation section accordingly (Line 679).

25. reference list needs to be proof read as some titles do not adhere to Vancouver style: e.g # 37,38,61,62,67

The reference list has been corrected.

We appreciate you taking the time to read and review our manuscript. 

Kind regards, 

Louise Carroll, Cliona O’ Sullivan, Catherine Doody, Carla Perrotta and Brona Fullen

---

## [Editor Report · Decision Letter 1]

14 Oct 2022

Pelvic Organ Prolapse: The Lived Experience

PONE-D-22-22443R1

Dear Dr. Carroll,

We’re pleased to inform you that your manuscript has been judged scientifically suitable for publication and will be formally accepted for publication once it meets all outstanding technical requirements.

Kind regards,

Antonio Simone Laganà, M.D., Ph.D.

Academic Editor

PLOS ONE

Additional Editor Comments (optional):

I carefully evaluated the revised version of this manuscript.

Authors have performed the required changes, improving significantly the quality of the paper.
---

## [Editor Report · Acceptance letter]

19 Oct 2022

PONE-D-22-22443R1 

Pelvic organ prolapse: The lived experience 

Dear Dr. Carroll:

I'm pleased to inform you that your manuscript has been deemed suitable for publication in PLOS ONE. Congratulations! Your manuscript is now with our production department. 

Kind regards, 

on behalf of

Dr. Antonio Simone Laganà 

Academic Editor

PLOS ONE